# Integrated Metabolomic and Transcriptomic Analysis Reveals the Flavonoid Regulatory Network by *Eutrema EsMYB90*

**DOI:** 10.3390/ijms22168751

**Published:** 2021-08-15

**Authors:** Yuting Qi, Chuanshun Li, Chonghao Duan, Caihong Gu, Quan Zhang

**Affiliations:** Shandong Provincial Key Laboratory of Plant Stress Research, College of Life Science, Shandong Normal University, Jinan 250014, China; qiyuting0430@163.com (Y.Q.); chuanshunli@163.com (C.L.); Duanchonghao1992@163.com (C.D.); gucaihong0217@163.com (C.G.)

**Keywords:** *Eutrema salsugineum*, *EsMYB90*, metabolites, metabolome, transcriptome, integrated analysis, flavonoid pathway

## Abstract

Flavonoids are representative secondary metabolites with different metabolic functions in plants. Previous study found that ectopic expression of *EsMYB90* from *Eutrema*
*salsugineum* could strongly increase anthocyanin content in transgenic tobacco via regulating the expression of anthocyanin biosynthesis genes. In the present research, metabolome analysis showed that there existed 130 significantly differential metabolites, of which 23 metabolites enhanced more than 1000 times in *EsMYB90* transgenic tobacco leaves relative to the control, and the top 10 of the increased metabolites included caffeic acid, cyanidin O-syringic acid, myricetin and naringin. A total of 50 markedly differential flavonoids including flavones (14), flavonols (13), flavone C-glycosides (9), flavanones (7), catechin derivatives (5), anthocyanins (1) and isoflavone (1) were identified, of which 46 metabolites were at a significantly enhanced level. Integrated analysis of metabolome and transcriptome revealed that ectopic expression of *EsMYB90* in transgenic tobacco leaves is highly associated with the prominent up-regulation of 16 flavonoid metabolites and the corresponding 42 flavonoid biosynthesis structure genes in phenylpropanoid/flavonoid pathways. Dual luciferase assay documented that EsMYB90 strongly activated the transcription of *NtANS* and *NtDFR* genes via improving their promoter activity in transiently expressed tobacco leaves, suggesting that EsMYB90 functions as a key regulator on anthocyanin and flavonoid biosynthesis. Taken together, the crucial regulatory role of *EsMYB90* on enhancing many flavonoid metabolite levels is clearly demonstrated via modulating flavonoid biosynthesis gene expression in the leaves of transgenic tobacco, which extends our understanding of the regulating mechanism of MYB transcription factor in the phenylpropanoid/flavonoid pathways and provides a new clue and tool for further investigation and genetic engineering of flavonoid metabolism in plants.

## 1. Introduction

Flavonoids, as the most ancient and vital secondary metabolites in plants, are determinants of crop quality and commercial value because they influence many characteristics of plant tissue including color, nutritive value, aroma and antioxidant properties [1]. More than 6000 different flavonoids widely distributed in plants have been identified [2,3]. Accumulation of flavonoids is favorable for plants because they not only provide plants with color for most flowers, fruits and seeds to attract pollinators, but also light-shielding, metal-binding and antioxidant capacities and functions in osmotic regulation [3,4,5]. A common factor with abiotic stress is the accumulation of ROS [6]. Flavonoids, acting as scavengers of ROS, can be induced by many environmental stimuli to protect plants against diverse abiotic and biotic stresses [6,7]. In general, flavonoids are classified into several subgroups including anthocyanin, proanthocyanidin (PA or condensed tannin), flavonol, flavone, flavanone and isoflavone according to their structures and modifications of the A, B and C rings [2,7,8]. The various classes of flavonoids are synthesized via the phenylpropanoid/flavonoid pathways [1,9]. In *Arabidopsis*, scaffold structures of flavonoids include kaempferol, quercetin and isorhamnetin for flavonols, cyanidin for anthocyanins and epicatechin for proanthocyanidins [8]. The number of flavonoid molecules increases enormously by modifying these scaffolds, including glycosylation, methylation and acylation, and these modified flavonoid products change their stability, solubility or localization [8,9]. Moreover, the structural diversity of flavonoids leading to the diversity of physicochemical and biological properties allow them to interact with targets in different subcellular locations to influence their biological activity in plants, animals and microbes [10,11]. Overall, the structural diversity and antioxidant properties of flavonoids contribute to their important roles in the stress defense of plants as well as the agronomic and nutritional value of plant products [2,3,7]. Thus, the enhancing the content of anthocyanins and flavonoids in food plants represents an important objective in crop genetic improvement, which also explains why the study of flavonoid metabolism has gained increasing interest in recent years [12,13,14].

The central pathway of flavonoid biosynthesis is conserved in plants [3]. So far, the enzymes involved in flavonoid biosynthesis have been well characterized, not only in *Arabidopsis*, but also in apple, grape (*Vitis vinifera*), pear (*Pyrus communis* L.), petunia (*Petunia hybrida*) and rose (*Rosa hybrida*) [15,16,17,18]. The conversion of phenylalanine to trans-cinnamic acid, catalyzed by phenylalanine ammonia lyase (PAL), is the initial step in the flavonoid pathway [15]. Chalcone synthase (CHS) is the first committed enzyme in the biosynthesis of all flavonoids [8]. Chalcone isomerase (CHI) catalyzes the stereospecific cyclization of naringenin chalcone to naringenin, followed by the naringenin forming dihydroflavonol (dihydrokaempferol) catalyzed by flavanone 3-hydroxylase (F3H) [8]. Flavonoid 3′-hydroxylase (F3′H), a cytochrome P450 monooxygenase, can convert dihydrokaempferol or kaempferol to dihydroquercetin and quercetin, respectively [8]. Early biosynthesis genes (EBGs) in flavonoid pathway encoding PAL, CHS, CHI and F3′ H, are involved in the production of common precursors of various flavonoids [1]. The late biosynthetic genes (LBGs) encoding dihydroflavonol reductase (DFR), leucoanthocyanidin dioxygenase/anthocyanidin synthase (LDOX/ANS), UDP-glucose:flavonoid 3-glucosyltransferase (UFGT), anthocyanidin reductase (ANR) and leucoanthocyanidin reductase (LAR) are specifically expressed for various flavonoid derivatives [1].

The first committed reaction leading to anthocyanin and PA is the reduction of dihydroflavonol to the corresponding leucoanthocyanidin catalyzed by DFR, an important branching point protein [8]. The first colored compound in the anthocyanin biosynthetic pathway is anthocyanidin, and LDOX/ANS catalyzes the formation of anthocyanidin from leucoanthocyanidin [8]. The glycosylation of anthocyanidin, catalyzed by UFGT to form anthocyanin, is essential for stable anthocyanin accumulation [8]. For example, flavonoid 3-O-glucosyltransferase (UGT78D2, At5g17050) can transfer glucose to the C-3 position of both flavonol and anthocyanin aglycones [19]. Cyanidin (Cy), delphinidin (Dp), pelargonidin (Pg), peonidin (Pn), petunidin (Pt) and malvidin (Mv) are six common anthocyanin derivatives [20]. Dp, Pn and Mv are sources of purple and dark colors, whereas Cy and Pg are the main pigments in bright red colors [20]. The synthesis of epicatechin (*cis*-flavan-3-ol) and catechin (*trans*-flavan-3-ol) as monomers of PAs are catalyzed by ANR and LAR, respectively [21]. PAs, mainly including catechin and epicatechin, are thought to be stored in the vacuoles of seed coat endothelial cells, and their accumulation could confer a brown hue of matured seeds [22,23]. Arabidopsis seeds accumulate only PA and flavonol that provide the protection of the embryo and endosperm [24]. Other seeds such as barley and sorghum also accumulate PA in the testa of the grain [7]. In grapevine (*Vitis vinifera*), anthocyanin and PA associated to properties of color, bitterness and astringency in red wine are the main flavonoids in fruits [21]. Roots of maize plants exposed to aluminum stress could exude high levels of catechin and quercetin, which indicates their ability of chelating metals to ameliorate aluminum toxicity [3,25].

Flavonols, having a wide range of potent physiological activities, are probably the most important flavonoids participating in stress responses [3,26]. The first committed step for biosynthesis of flavonol branching is catalyzed by flavonol synthase (FLS) [27]. Flavonol, PA and anthocyanin share several biosynthetic steps in the flavonoid pathway [15]. Dihydroflavonols as the competing substrates for DFR, FLS, F3′H and F3′5′H are the precursors of both anthocyanin and flavonol [28]. In *Arabidopsis*, leaves, stems, flowers and seedlings mainly accumulate kaempferol glycosides, whereas in mature seeds, quercetin-3-O-rhamnoside (quercitrin) is the major flavonol [7]. Additionally, flavone and isoflavone are synthesized at a branch point of the anthocyanin/PA pathway using flavanone as the direct precursor [1].

In diverse higher plant species, the flavonoid biosynthesis pathway is transcriptionally regulated by a conserved MYB-bHLH-WD40 (MBW) complex [1,2,9]. Several R2R3-MYB family members have been shown to separately control flavonoid biosynthesis through various flavonoid pathway branches [29,30]. The flavonol regulators MYB12/PFG1 (production of flavonol glycosides1), MYB11/PFG2 and MYB111/PFG3 regulate the expression of early biosynthetic genes encoding CHS, CHI, F3H, F3′H and FLS, whereas the anthocyanin regulators MYB75/PAP1 (production of anthocyanin pigment1), MYB90/PAP2, MYB113 and MYB114 control the expression of the late biosynthetic genes encoding DFR and LDOX/ANS [6,29,31,32]. The sequence diversity of R2R3-MYB causes differences in anthocyanin and flavonoid accumulation [33]. When *AtMYB75* was stably expressed in the hop genome, the female flowers and cones of transgenic plants were reddish to pink, and higher levels of anthocyanin, rutin, isoquercitin, kaempferol-glucoside and kaempferol-glucoside-malonate were also detected in transgenic plants compared to wild type plants [34].

The level of metabolites as the end product of gene expression can be regarded as the ultimate response of biological systems to genetic or environmental changes, and metabolomics describes the metabolite changes that occur in organisms [35,36]. Flavonoids, a representative group of secondary metabolites, have roles in many facets of plant physiology, including the well-known antioxidant and health-promoting effects of these compounds [6]. Salt cress (*Eutrema salsugineum*), a *Brassicaceae* species closely related to *Arabidopsis*, is an important model halophyte. Tobacco, as a model plant for genetic transformation, is very convenient for transformation of genes and molecular biology research. In our previous research, we have successfully obtained *Eutrema EsMYB90* transgenic tobacco plants [35]. Here, the function of EsMYB90, an R2R3-MYB transcription factor on regulating biosynthesis of flavonoid metabolites, was investigated by the integrated analysis of metabolite profiling and RNA-seq as well as the dual luciferase assay in the leaves of *EsMYB90* transgenic tobacco in comparison with wild type plants. The result manifested that the *EsMYB90* gene is responsible for the accumulation of flavonoids and other antioxidant metabolites, and *EsMYB90* plays a pivotal regulating role in flavonoid metabolite biosynthesis via modulating the expression of flavonoid biosynthesis genes such as *NtANS* and *NtDFR*. Therefore, the study exhibited a growing insight for us to understand and genetically manipulate the biosynthesis of anthocyanin and flavonoids to generate a novelty of colors and enhance antioxidant activity in plants.

## 2. Results

### 2.1. Increased Flavonoid Compounds in EsMYB90 Transgenic Tobacco Leaves

In our previous study, we demonstrated that ectopic expression of *Eutrema*
*EsMYB90* (*p35S:EsMYB90*) in tobacco led to purplish red at varying degrees in leaves, stem, flowers and fruit pods and was accompanied with a strongly elevated anthocyanin level [37]. To further understand and define the regulating effect of the *EsMYB90* gene on different pigments and flavonoid compounds, the contents of anthocyanin, chlorophyll, proanthocyanin (PA) and total flavonoid were checked in *EsMYB90* transgenic tobacco and wild type plants. We found that the contents of total flavonoid and anthocyanin in young leaves (YL), mature leaves (ML), stems and roots in *EsMYB90* transgenic tobacco L2 and L4 lines were extremely higher than that of wild type. Moreover, the content of anthocyanin, total flavonoid and proanthocyanidin in the tissues detected from L4 EsMYB90 transgenic tobacco line were all significantly higher than that of the L2 line, except for total flavonoid content in young leaves (YL). The contents of anthocyanin and total flavonoid in the ML of transgenic L4 line were respectively elevated more than 979- and 591-fold (Figure 1A,B, Appendix A). Additionally, in the mature leaves of transgenic lines L2 and L4, PA levels were significantly up-regulated, whereas chlorophyll content almost remained unaffected (Figure 1C,D and Appendix A).

Collectively, the results demonstrate that ectopic expression of the *EsMYB90* gene not only led to the markedly higher accumulation of anthocyanin but also the contents of total flavonoid and PA in transgenic tobacco.

### 2.2. Metabolite Analysis Based on OPLS-DA Model

In order to preliminarily understand the overall metabolic differences among samples and the degree of variability between samples within groups, PCA analysis was performed for all samples in wild type (WT) and *EsMYB90* transgenic tobacco (T). From the analysis result, it can be observed that the WT and T of inter-groups were clearly separated, and the repeated samples in intra-groups were gathered together in the PCA score plot, thus indicating that the experiment was reproducible and reliable (Appendix A).

Q^2^ is an important parameter for evaluating the models of OPLS-DA, and the value of Q^2^ ≥ 0.9 indicates OPLS-DA as an excellent model. The OPLS-DA model in our study showed that Q^2^ values of WT and T of inter-groups were equal to 0.994, R^2^X = 0.863 and R^2^Y = 1, which demonstrated that the model was reliable and meaningful (Appendix A). The VIP value indicates the importance of a variable for the entire model, and a variable with a VIP greater than 1 is regarded as responsible for separation [38]. In this study, 130 discriminating metabolites were screened and distinguished by the combining fold change ≥ 2 (or ≤0.5) with VIP ≥ 1 in the OPLS-DA model.

### 2.3. Metabolic Profiling and Significantly Differential Metabolite Analysis

To explore the functions of *EsMYB90* genes in modulating flavonoid biosynthesis, we investigate the differential metabolites in leaves of *EsMYB90* transgenic tobacco relative to wild type. A total of 550 metabolites obtained from UPLC-MS/MS (ultra-performance liquid chromatography-tandem mass spectrometry) were analyzed in leaves using a widely targeted metabolomics method (Figure 2A and Appendix A). Combining FC (fold change) ≥2 (or ≤0.5) and VIP ≥ 1 in OPLS-DA model, 130 significantly differential metabolites including 50 flavonoids, 9 hydroxycinnamoyl derivatives, 13 quinate derivatives and 3 coumarins were screened and identified (Figure 2 and Appendix A). Among these, 50 significantly differential flavonoid metabolites were divided into 7 categories including anthocyanin (1), flavone (14), flavonol (13), flavone C-glycosides (9), flavanone (7), isoflavone (1) and catechin derivatives (5) (Figure 2B, Appendix A). In the 50 significantly differential flavonoid metabolites, contents of 46 metabolites were increased more than twofold in transgenic tobacco leaves, of which 15 metabolites with the negligible expression levels were observed in leaves of wild type plants (Appendix A). Above all, the most prominently differential metabolites were cyanidin O-syringic acid in anthocyanin, followed by myricetin in flavonol and chrysoeriol 5-O-hexoside in flavone (Figure 2A and Appendix A).

Except for the above flavonoid metabolites, the contents of caffeic acid in hydroxycinnamoyl derivatives, 3-O-p-coumaroyl shikimic acid O-hexoside as well as chlorogenic acid methyl ester in quinate derivatives, esculetin (6,7-dihydroxycoumarin) and esculetin O-quinacyl esculetin O-quinic acid in coumarin also strongly increased (Figure 2A and Appendix A). The hydroxycinnamoyl derivative, quinate derivative and coumarin possess a wide range of antibacterial and antiviral activities and the capacity of absorbing ultraviolet rays; thus, these metabolites also possibly play an important role in improving the antioxidative activity of plants.

In summary, the level of flavonoid metabolites such as cyanidin O-syringic acid, myricetin and chrysoeriol 5-O-hexoside and other metabolites including caffeic acid, chlorogenic acid methyl ester and esculetin were all strikingly up-regulated, which suggested that the *Eutrema EsMYB90* transcription factor plays a critical regulating role on the enhanced level of antioxidant metabolites in transgenic plants.

### 2.4. Integrated Analysis of Metabolite Profiling and RNA-seq in Phenylpropanoid/Flavonoid Biosynthesis Pathways

The increased anthocyanin content in leaves, stems, flowers and fruit pods of *EsMYB90* transgenic tobacco plants was caused by significantly up-regulated expression of anthocyanin biosynthesis genes in the previous study [37]. In order to clarify in depth the specific roles of *EsMYB90* in flavonoid pathways and unravel the regulatory mechanisms in place, we have performed the combined analysis of significantly differential metabolites and markedly differential expression genes (DEGs). This result showed that 16 markedly up-regulated metabolites and 15 key enzymes encoded by 53 significantly DEGs were mapped onto the phenylpropanoid/flavonoid metabolism pathways (ko00940, ko00941, ko00942, ko00943 and ko00944), as shown in Figure 3 (Appendix A). In the 53 DEGs in phenylpropanoid/flavonoid pathways, the expression of 42 genes was strongly up-regulated, including all genes that encoded PAL (phenylalanine ammonia-lyase), C4H (cinnamate-4-hydroxylase), CCoAOMT (caffeoyl-CoA O-methyltransferase), CHI (chalcone isomerase), F3′H (flavonoid 3′-monooxygenase), DFR (dihydroflavonol 4-reductase), ANS/LDOX (anthocyanidin synthase/leucoanthocyanin dioxygenase), 3GT/BZ1 (anthocyanidin 3-O-glucosyltransferase) and UFGT (UDP-glucose: flavonoid 3-O-glucosyltranferase) (Appendix A).

In the phenylpropanoid pathway (ko00940), the DEGs encoding PAL (107802063, 107761482 and 107820497, 107769293) were up-regulated more than threefold, and the level of corresponding metabolites of cinnamic acid and caffeic acid markedly increased (Figure 3, Appendix A). In the flavonoid biosynthesis pathway (ko00941), the 42 DEGs and 12 differential metabolites were shown as black arrow lines in a map of the integrated metabolic pathways. In detail, 12 differential metabolites were up-regulated more than 10 times, including caffeoyl quinic acid in the quinate derivatives; naringin, hesperetin, eriodictyol, homoeriodictyol and prunin in the flavanones; myricetin, dihydrokaempferol (DHK) and dihydroquercetin (DHQ) in the flavonols; tricetin in the flavones; and epigallocatechin and L-epicatechin in the catechin derivatives. Of the 42 DEGs, 32 genes encoding C4H (3), HCT (6), CCoAOMT (2), CHS (4), CHI (4), F3H (2), F3′H(2), DFR (2), ANS/FLS (2) and ANS/ LDOX (5) were markedly up-regulated (Figure 3, Appendix A). 2′-hydroxydaidzein in isoflavonoid biosynthesis pathways (ko00943) and isotrifoliin in flavone and flavonol pathways (ko00944) were enhanced more than twofold, while the expression of genes encoding F3′H (2) and 3GT (2) that correspond to these metabolites’ biosynthesis are also significantly up-regulated. It is noteworthy that F3′H (2) and 3GT (2) also exist in the flavonoid biosynthesis pathway (ko00941) and anthocyanin biosynthesis pathway (ko00942), respectively (Figure 3 and Appendix A).

In addition, cyanidin O-syringic acid, the only differential metabolite detected in anthocyanin but presently not annotated in the KEGG pathway, had a dramatically higher level in the leaves of *EsMYB90* transgenic tobacco when compared to the wild type (Figure 3, Appendix A). The specific enzymes for synthesizing anthocyanin are DFR, ANS, 3GT/BZ1 and UFGT [39]. In our study, all genes encoding DFR(2), ANS/LDOX (5), ANS/FLS (2), 3GT/BZ1(2) and UFGT (4) were markedly up-regulated (Figure 3, Appendix A); thus, these up-regulated DEGs are possibly responsible for the strongly enhanced cyanidin O-syringic acid content in the leaves of *EsMYB90* transgenic tobacco.

Taken together, these markedly increased flavonoid metabolites were tightly associated with the up-regulated expression of corresponding genes controlling their biosynthesis in flavonoid/phenylpropanoid pathways.

### 2.5. EsMYB90 Enhanced Flavonoid Metabolite Level via Activating Transcription of Flavonoid Biosynthesis Genes

Anthocyanidin synthase (ANS) and dihydroflavonol 4-reductase (DFR) encoding genes have been widely characterized as the key anthocyanin biosynthetic genes [28]. In order to expound the regulatory mechanism of *EsMYB90* genes on biosynthesis of anthocyanin and flavonoids, firstly, the MYB-binding motif analysis in promoters of *NtANS* and *NtDFR* genes were analyzed using PlantCARE and visualized by Tbtools software. The result showed that five MYB transcript factor binding elements were found in the promoter of *NtANS*, while six MYB-binding elements existed in the promoter of *NtDFR* (Figure 4A,B, Appendix A). Further, the *Agrobacterium* containing *pCAMBIA3301H-p35S:EsMYB90* as an effector as well as the *Agrobacterium* containing *pGreenII0800-pNtANS:LUC* or *pGreenII0800-pNtDFR:LUC* as the reporter were co-transfected into tobacco leaves, and the dual luciferase assay was performed. The results demonstrated that the relative activity ratio of the pNtANS:LUC firefly luciferase to renilla luciferase (p35S:REN) was up-regulated about four times (Figure 4C, Appendix A), while the ratio of pNtDFR:LUC firefly luciferase to renilla luciferase (p35S:REN) was elevated 23 times (Figure 4D, Appendix A). This study documented that EsMYB90 TF could operate as a positive transcriptional regulator of *NtANS* and *NtDFR* genes by directly binding to the MYB-binding elements of their promoters.

To investigate the different kinds of metabolites regulated by *EsMYB90* in place, the significant correlations were analyzed between the *EsMYB90* gene and the differential metabolites. On account of the screening criteria required for Pearson correlation coefficient |PCC| ≥ 0.8 and *p*-value of correlation (PCCP) ≤ 0.05, 80 metabolites content were significantly influenced by *EsMYB90*, including 35 flavonoid metabolites and 45 other metabolites (Figure 4E, Appendix A). The 35 flavonoid metabolites consist of 10 flavone, 9 flavonol, 7 flavanone, 5 flavone C-glycosides, 3 catechin derivatives and one anthocyanin (cyanidin O-syringic acid), while 45 other metabolites mainly include 8 quinate derivatives, 6 hydroxycinnamoyl derivatives and 2 coumarins (Figure 4E, Appendix A). The results suggested that *EsMYB90* expression in transgenic tobacco could significantly enhance the biosynthesis of flavonoids and other antioxidant metabolites.

To further identify the DEGs and differential metabolites affected by *EsMYB90* in transgenic tobacco leaves in phenylpropanoid/flavonoid pathways, we constructed a regulating network between the 15 significantly differential metabolites (14 up-regulated and one down-regulated) and the DEGs related to these metabolites in the ko00940, ko00941, ko00943 and ko00944 pathways. The results displayed that 424 significant correlations were established between 36 DEGs and 15 markedly differential metabolites on the basis of the criteria of |PCC| ≥ 0.8 and PCCP ≤ 0.05 (Figure 4F, Appendix A). Moreover, except for L-epicatechin and medicarpin which only correlated with *HCT* (107830064) and *IF7MAT* (107782061) genes, each of the other 13 metabolites was correlated with more than 27 DEGs. Further, in the 15 metabolites, except for caffeic acid (ko00940), medicarpin (ko00943) and isotrifoliin (ko00944), the other 12 metabolites all were enriched in the flavonoid biosynthesis pathway (ko00941), including 5 flavanones (naringin, hesperetin, eriodictyol, homoeriodictyol, prunin), 3 flavonols (myricetin, dihydrokaempferol, dihydroquercetin), 2 PAs (epigallocatechin, L-epicatechin), one flavone (tricetin) and one quinate derivative (chlorogenic acid). Moreover, 277 Pearson correlations were formed between the 12 up-regulated flavonoid metabolites and 27 DEGs (ko00941).

In summary, *EsMYB90* could strikingly increase the level of flavonoid metabolites via modulating the flavonoid biosynthesis genes in transgenic tobacco leaves, indicating that *EsMYB90* plays crucial roles in improving antioxidant capacity by strongly elevating the level of flavonoid metabolites in transgenic plants.

## 3. Discussion

### 3.1. Key Role of EsMYB90 in Enhancing Antioxidative Metabolite Level

Anthocyanins are best known for their function in the color of fruits, flowers and autumn leaves [5,40,41]. In the present study, the content of anthocyanin and total flavonoid in young leaves (YL), mature leaves (ML), stems and roots in *EsMYB90* transgenic tobacco were extremely higher than that of wild type (Figure 1A,B, Appendix A). Further, the metabolome analysis showed that a dramatically higher level of cyanidin O-syringic acid, as the sole significantly differential metabolite in anthocyanin, was found in leaves of *EsMYB90* transgenic tobacco, while a negligible level was found in wild type plants (Figure 2 and Appendix A). Meanwhile, cyanidin, rosinidin O-hexoside, malvidin 3-O-glucoside (oenin), malvidin 3-O-galactoside and delphinidin 3-O-glucoside (mirtillin) in anthocyanins were detected with unchanged level amounts in leaves of transgenic tobacco relative to wild type (Appendix A). Thus, it is proposed that cyanidin O-syringic acid is a crucial pigment compound for the purplish red phenotype in the leaves of *EsMYB90* transgenic tobacco. Flavonol was reported to participate in stress responses and is involved in complex interplay between flavonols and ROS [7]. Dihydroflavonols including DHK (dihydrokaempferol), DHQ (dihydroquercetin) and DHM (dihydromyricetin) are precursors of anthocyanins and flavonols and could further form unstable anthocyanins via catalysis of DFR and ANS [28]. There were 13 flavonol metabolites which increased more than six times in the leaves of transgenic tobacco compared with wild type plants, including myricetin, kaempferol-3-O-robinoside-7-O-rhamnoside (robinin), DHK, quercetin 3-alpha-L-arabinofuranoside (avicularin), quercetin O-acetylhexoside, quercetin 4′-O-glucoside (spiraeoside), isorhamnetin O-acetyl-hexoside and DHQ (Figure 5, Appendix A). PAs contribute to the protection of plants against abiotic stresses and microbial attacks [42]. In our study, epigallocatechin (EGC), L-epicatechin and protocatechuic acid in PAs all increased more than 50 times (Figure 5a and Appendix A). In addition, 13 of 14 flavones, 7 of 9 flavone C-glycosides, 7 flavanones and one isoflavone were all significantly up-regulated in the leaves of *EsMYB90* transgenic tobacco relative to wild type (Figure 5 and Appendix A). Therefore, it is inferred that these flavonols, PAs, flavones, flavone C-glycosides, flavanones and isoflavone up-regulated by *EsMYB90* were the important flavonoid metabolites for improving the antioxidant capacity of plants.

Flavonoids play a prominent role in many stress responses owing to their functions in protecting the photosynthetic machinery of plants exposed to an excess of light (especially UV-B), scavenging reactive compounds including ROS occurring under various stress conditions, binding various heavy metals and sequestering the toxic elements [43], exhibiting the osmotic adjustment to remain turgid in plant cells under low water availability and enhancing the efficiency of nutrient retrieval during senescence [3,5,44,45,46,47]. In summary, a diverse array of flavonoids in higher plants have evolved according to their physiological and ecological functions to adapt to various environmental pressures [8]. Moreover, flavonoids are also beneficial to the agronomic value of plant products and the health value of food [2,20]. A series of flavonoid subgroups such as PAs, anthocyanins, flavonols, flavones, flavanones and isoflavones have been recognized in over 500 food items by the US Department of Agriculture as dietary flavonoids with consistent evidence of beneficial effects in humans [6]. In this study, the 92 metabolites including 46 flavonoids, 6 hydroxycinnamoyl derivatives, 12 quinate derivatives and 3 coumarins were markedly up-regulated in 130 significantly different metabolites, and most of them have strongly antioxidant capacities (Figure 2 and Appendix A).

Our transcriptomic analysis herein revealed that many of the genes involved in phenylpropanoid/flavonoid metabolism, from *PAL* to *UFGT* genes, can be directly or indirectly up-regulated by the EsMYB90 transcription factor. Moreover, the transcript abundance of flavonoid biosynthesis genes coincided with the corresponding metabolite levels detected (Figure 3). Significantly, DFR, ANS and UFGT are the crucial structural proteins that facilitate anthocyanin pigmentation in plants. In this study, *EsMYB90* strongly activated the flux of flavonoid intermediates toward the production of cyanidin O-syringic acid in anthocyanin by elevating the transcription of *DFR*, *ANS* and *UFGT* genes, with a resulting purple-red leaf phenotype (Figure 1, Appendix A). Meanwhile, the level of epicatechin gallate (ECG) and L-epicatechin in PAs were markedly enhanced, which is in accordance with the significantly up-regulated expression of *DFR* and *ANS* genes (Figure 5, Appendix A). Additionally, strikingly up-regulated differential metabolites including myricetin in flavonol, chrysoeriol 5-O-hexoside in flavone and naringin in flavanone (Figure 5, Appendix A) indicated that *EsMYB90* also strongly modulated the flux of flavonoid intermediates toward the production of flavonols, flavones and flavanones. An exception is the down-regulated expression of the leucoanthocyanidin reductase encoding gene (*LAR*) in *EsMYB90* transgenic tobacco leaves (Figure 3 and Appendix A), suggesting the existence of a different regulatory mechanism controlling the various flavonoid metabolite biosyntheses via regulating the expression of a particular gene.

Thus, it is reasonable to suggest that the tobacco plants overexpressing the *EsMYB90* gene alone showed similar changes in levels of both metabolites and transcripts, which supports that EsMYB90 is fulfilling a key central role by regulating the most biosynthetic genes related to the phenylpropanoid/flavonoid pathways in transgenic tobacco leaves.

### 3.2. Regulating Mechanism of EsMYB90 in Phenylpropanoid/Flavonoid Pathways and Its Novelty

Several R2R3 MYB transcription factors are known to regulate the branches of the phenylpropanoid/flavonoid metabolic networks, and various end-products of the associated pathways accumulate in specific cells, tissues and organs in higher plants [30]. *Arabidopsis*, *AtMYB75*, *AtMYB90*, *AtMYB113*, *AtMYB114* and *AtMYBL2* act as crucial modulators of flavonoid biosynthesis [32,48,49,50]. Furthermore, *AtMYB75*/*PAP1* is a master regulator of the flavonoid/anthocyanin biosynthesis pathway, and *AtMYB75*/*PAP1* and *MYB90*/*PAP2* can work in a combinatorial way with TTG1, a WD40 protein, and different bHLH partners such as TT8, GL3 or EGL3 [7,14,51]. On the other hand, AtMYB60 is the first MYB protein identified that functions as a transcriptional repressor in anthocyanin biosynthesis [30]. Thus, it can be seen that the MYB transcription factor, acting as activator or repressor, plays a critical role in the modulation of flavonoid biosynthesis [30].

In *Arabidopsis*, *AtMYB75*, *AtMYB90*, *AtMYB113* and *AtMYB114* mainly regulate the expression of late biosynthesis genes (LBGs) [5,32]. However, *Eutrema EsMYB90* is not only involved in regulating the expression of LBGs (*DFR*, *ANS*/*LDOX* and *UFGT*) but also in promoting the expression of anthocyanin early biosynthesis genes (EBGs) such as *PAL*, *CHS*, *CHI* and *F3H* in *35S:EsMYB90* tobacco transgenic plants [37]. Additionally, *Arabidopsis AtMYB75*, *AtMYB90*, *AtMYB113* and *AtMYB114* function redundantly in participating in the MYB-bHLH-WD40 (MBW) complex to regulate anthocyanin and proanthocyanidin (PA) biosynthesis, and at least four MBW complexes assembled with various MYBs were involved in PA accumulation of the innermost cell layer in an *Arabidopsis* seed coat [2,52]. EsMYB90 protein has the closest homologous relationship with EsMYB5, EsMYB15, EsMYB106, EsMYB108 and EsMYB-related protein 340 in *Eutrema*; however, the ANDV and KPRPR [S/T] F motifs characterizing anthocyanin-promoting MYBs only exist in EsMYB90 protein, but not in the other five EsMYBs, while the conserved [D/E]LX_2_[R/K]X_3_LX_6_LX_3_R motif for interacting with an R/B-like bHLH protein only exists in EsMYB90 and EsMYB5 proteins [37]. Thus, it was suggested that EsMYB90 is a major player in flavonoid biosynthesis in *Eutrema*, which is different from that in *Arabidopsis* [37].

In view of the targets of MYB TFs suggested to be different in different plant species, the choice of gene source is important for an efficient change of the metabolites in the flavonoid pathways [3,53]. Moreover, understanding how phenylpropanoid/flavonoid metabolism changes can be regulated by the MYB transcription factor would be useful for improving crop yield and quality via enhancing the antioxidative capacity of plants. Therefore, this study was designed to evaluate the influence of halophyte *Eutrema EsMYB90* on the phenylpropanoid/flavonoid pathways and potential regulatory mechanisms in transgenic plants. In this study, the combined analysis of transcriptome and metabolome as well as a dual luciferase assay were employed to clarify the regulatory mechanism of *EsMYB90* on flavonoid biosynthesis and verify the function of *EsMYB90* on metabolite biosynthesis. The result found that ectopic expression of the *EsMYB90* gene in transgenic tobacco increased 46 flavonoid levels as well as the content of 6 hydroxycinnamoyl derivatives, 12 quinate derivatives and 3 coumarins with the antibacterial and antiviral activities (Figure 2, Appendix A). Moreover, the evidence from our experiments demonstrated that the flavonoid biosynthetic pathways in transgenic tobacco can be switched on by *EsMYB90*, which will help to deepen our understanding of plant flavonoid metabolite biosynthesis. Thus, it is inferred that the *EsMYB90* gene, as a novel activator of flavonoid biosynthesis, possibly acts effectively in most plant species to promote flavonoid levels.

### 3.3. EsMYB90—A Potential Important Gene for Genetic Breeding to Improve the Flavonoid Level of Crops

In the last several decades, the majority of flavonoid molecules and the genes involved in biosynthesis of these diverse compounds in *Arabidopsis* have been intensively identified [54]. Moreover, a huge number of genetic engineering attempts have been described to produce novel flower colors in several plant species, such as petunia, gerbera, rose and carnation, by modifying the anthocyanin biosynthesis pathway [55,56,57]. However, the diversified functionality of various flavonoids and the transcription regulation of the enormous diversification of flavonoids should be addressed by further investigation.

In higher plants, flavonoid and anthocyanin biosynthesis are regulated by a conserved MYB-bHLH-WD40 (MBW) complex, and the different sets of MBW complexes exerted in different plants [2]. Of these, R2R3-MYB plays pivotal role in transcriptional regulation of flavonoid biosynthesis. EsMYB90 protein, an R2R3-MYB from *Eutrema*, has a conserved DNA-binding domain (R2 and R3 repeats) in the N-terminal, and a conserved [D/E]LX_2_[R/K]X_3_LX_6_LX_3_R motif required for interaction with R/B-like bHLH proteins [37,57]. The phylogenetic tree of EsMYB90 with 29 MYB proteins involved in flavonoid synthesis in 16 plants demonstrated that EsMYB90 has a closer relationship with *Arabidopsis* AtMYB75, AtMYB90, AtMYB113, AtMYB114, *Brassica oleracea* BoMYB1 and *Brassica rapa* BrMYB114 [37]. Further, the ANDV motif, a characteristic identifier for anthocyanin-promoting MYBs in dicots, exists in EsMYB90, AtMYB90, AtMYB75, AtMYB113 and AtMYB114 [37,58]. However, EsMYB90 protein exists only at 80.5%, 78.9%, 78.4%, 74.4%, 65.9%, and 50% identities respectively in BoMYB1, AtMYB90, BrMYB114, AtMYB75, AtMYB113 and AtMYB114 [37]. In our current findings, the direct activation of *NtANS* and *NtDFR* exerted by EsMYB90 protein resulted in a phenotypic effect of anthocyanin accumulation in transgenic tobacco plants. Furthermore, we demonstrated that the critical regulation function of the *Eutrema EsMYB90* gene on the flavonoid regulatory network allowed the significant accumulation of 46 flavonoid compounds. These findings could be the basis for further engineering of flavonoids and optimization of the metabolic pathway. Biotechnological applications for the promotion of human health by engineering of flavonoids is a promising area [59,60]. Considering various flavonoids not only provide the abundant colors of plants, but also possess extensive antioxidative effects, antiviral activities and neuroprotective properties, their accumulation is a key objective for the genetic improvement of crops [3,61,62]. All in all, the EsMYB90 transcription factor on the regulatory network involved in several already annotated flavonoid biosynthesis genes and flavonoid metabolites, as well as its conservation of structure, revealed that the *EsMYB90* gene is a potential excellent gene for genetic breeding to enhance the flavonoid level of crops, and the study of modifying flavonoid content by *Eutrema EsMYB90* in plant tissues also opens up a new avenue to improve the agricultural and economic performance of crops.

This study describes the importance of flavonoid, and the critical role of *EsMYB90* gene in controlling the flavonoid biosynthesis in transgenic tobacco. However, the functional conservation of *EsMYB90* gene needs to be explored and studied furtherly to evaluate its application prospect in some important ecnomic crops.

## 4. Materials and Methods

### 4.1. Plant Materials and Growth Conditions

In our previous research, eighteen *35S:EsMYB90* transgenic tobacco lines were obtained, and almost all transgenic tobacco lines exhibited purple-red leaves and corollas, purple-black sepals and fruit pods, whereas the leaves, sepals and fruit pods of wild type were green, with pink corollas. Specifically, L2 and L4 transgenic lines displayed more obvious pigmentation [37]. In the research, L2 and L4 transgenic lines were used to determine the content of anthocyanin, total flavonoid, proanthocyanidin and chlorophyll as well as L4 lines for subsequent analysis of metabolome and transcriptome.

Tobacco (*Nicotiana tabacum*) were grown in a mixture of vermiculite, perlite and peat moss (1:1:1) at a greenhouse with a temperature of 25 °C and a photoperiod of 16 h light/8 h dark. After 6 weeks’ growth, the samples from the sixth leaves of wild type and transgenic tobacco plants at the period of 8–9 leaves, respectively, were collected for sequencing of transcriptome and metabolome. All samples were frozen immediately in liquid nitrogen and stored at −80 °C. The leaves from six individual plants were sampled as one biological replicate, and three biological replicates were used in this study.

### 4.2. Measurement of Anthocyanin, Total Flavonoid, PA and Chlorophyll

The young leaves (YL), mature leaves (ML), stems and roots of tobacco from six different plants that grew for about 8 weeks were respectively collected to quantify the anthocyanin, total flavonoid, PA and chlorophyll.

Anthocyanin and total flavonoid were extracted and quantified using the method described by Neff and Chory with minor modifications [63,64]. The absorbance of the samples was determined at 530 nm, 657 nm and 535 nm with a spectrophotometer (UV-1800, Shimadzu), and methanol with 1% HCl was used as a blank control. The anthocyanin content (mg/g FW) was calculated based on the formula of (A_530_ − 0.25 ×A_657_)/fresh weight (g), while the total flavonoid content was measured with (mg/g FW) = (1/958 × A_535_ × 10000 × V) /fresh weight (g), where the V indicates the total volume of the extract (mL).

For the determination of chlorophyll content, 80% acetone was used as the blank control, the absorbance of samples was detected at 663 nm and 645 nm and the chlorophyll content (mg/g FW) was calculated by the eq. (20.21× A_645_ + 8.02 × A_663_) / fresh weight (g) [65]. For the determination of proanthocyanidin content, 60% ethanol was used as the blank control, the absorbance of catechin standard samples were detected at 500 nm to draw the standard curve, and the PA content of each sample was calculated on the basis of the above standard curve using the improved vanillin colorimetric method [66]. At least three biological replicates were performed for each sample.

### 4.3. RNA-Seq and Bioinformatic Analysis

Total RNAs from the sixth leaves of wild type and EsMYB90 transgenic tobacco at the 7–8 leaf stage were isolated using a Quick RNA isolation kit (Bioteke Corporation, Beijing, China). The RNA library construction and sequencing were performed in the BGI Corporation (Shenzhen, China) using the BGISEQ-500 platform. Three independent biological replicates were carried out for the leaves of wild type and *EsMYB90* transgenic tobacco, respectively. The methods of getting the clean reads, gene functional annotation, identification of differential expressed genes, and GO and KEGG pathway functional enrichment analysis has been reported and shown in our previous publication [37]. The DEGs with log_2_Fold Change ≥1 (or ≤−1) and Padj ≤ 0.05 were designated as significantly differentially expressed genes.

### 4.4. Analysis of Metabolite Profiling

The qualitative and quantitative analysis of metabolites from tobacco leaf samples were performed using ultra performance liquid chromatography (UPLC) (Shim-pack UFLC SHIMADZU CBM30A, http://www.shimadzu.com.cn/) (Accessed on June 2018) and tandem mass spectrometry (MS/MS) (Applied Biosystems 6500 QTRAP, http://www.appliedbiosystems.com.cn/) (Accessed on June 2018) in MetWare (Wuhan, China). The mass spectrometry data was processed using the analyst 1.6.1 software. The metabolites were identified and analyzed based on the metabolite information public database and the local MetWare database MWDB (Metware Biotechnology Co., Ltd. Wuhan, China).

### 4.5. Differential Metabolite Analysis and Metabolic Pathways Construction

Principal component analysis (PCA), a multivariate technique for extracting the important information in inter- and intra-groups, was tested for the leaves of wild type (WT) and transgenic tobacco (T) to get the overall metabolic differences among samples and the degree of variability in samples within groups. Orthogonal partial least squares-discriminant analysis (OPLS-DA) is an effective method for screening differential metabolites to maximize the difference between groups. Q^2^ as an important parameter is used to evaluate the reliability of OPLS-DA model, and the value of Q^2^ ≥ 0.9 representing OPLS-DA is an outstanding model. In the study, the Q^2^ value of the OPLS-DA model is applied to distinguish the discriminated metabolites between samples of T and WT groups. Variable importance in projection (VIP), the weighted sum of squares of the OPLS- DA analysis, indicates the importance of a variable to the entire model [38]. In the study, the differential metabolites were screened by fold change ≥ 2 (or ≤0.5) and VIP ≥ 1.

The metabolomics data analysis was performed by MetaboAnalyst, and the metabolic pathways were constructed according to KEGG metabolic database.

### 4.6. Integrated Analysis of Metabolome and Transcriptome

For the combined analysis of metabolome and transcriptome, COR program from R was used to calculate the value of the Pearson correlation coefficient (PCC) in this study. In order to adequately define the networks of the *EsMYB90* gene regulating different kinds of metabolites via activating the transcription of flavonoid biosynthesis genes, we performed the analysis of significant correlation between: (1) the *EsMYB90* gene and the differential metabolites, (2) the differential expression genes (DEGs) and the differential metabolites in the phenylpropanoid/flavonoid pathways, based on the screening criteria required for the absolute value of the Pearson correlation coefficient (|PCC|) ≥ 0.8 and the *p*-value of correlation (PCCP) ≤ 0.05. The corresponding correlation network analysis was visualized with the Cytoscape software (version 3.3.0).

### 4.7. Dual Luciferase Assay

The conserved MYB-binding cis-element motifs located in the promoters of *NtANS* and *NtDFR* (named *pNtANS* and *pNtDFR*, respectively) were scanned by online software PlantCARE (http://bioinformatics.psb.ugent.be/webtools/plantcare/html/) (accessed on April 2021).

Dual luciferase assay has been widely applied to measure the TF (transcription factor)–promoter interactions to validate the transcriptional regulatory roles of TFs on target promoters. Here, the promoters (−2000 to 0) of *NtANS* (107819370) and *NtDFR* (107803097) genes were respectively cloned and recombined into the MCS (multiple cloning sites) of *pGreenII0800-LUC* as a reporter vector. The ORF of *EsMYB90* was inserted into the *Bam*HI and *Eco*RI sites of *pCAMBIA3301H* vector to get *pCAMBIA3301H-p35S:EsMYB90* as the effector vector, and the empty *pCAMBIA3301H* vector was used as a negative control. Primers for these constructs are listed in Appendix A. Then, all these constructs were individually were transferred into *Agrobacterium tumefaciens* GV3101. The GV3101 containing the *pGreenII0800-pNtANS* (or *pNtDFR*):*LUC* reporter gene vector was co-transfected with GV3101 containing the *pCAMBIA3301H-p35S:EsMYB90* effector vector into the leaves of tobacco (*Nicotiana benthamiana*). After 2–3 days of the tobacco leaves being transfected, we used the dual-luciferase reporter assay system (Meilunbio) to detect the enzyme activities of firefly luciferase (*pNtANS*:*LUC* or *pNtDFR*:*LUC*) and renilla luciferase (*p35S*:*REN*), respectively, in a highly sensitive plate chemiluminescence instrument (Centro LB960, Berthold Technologies, wildbad, Jerman). The relative luciferase activity (LUC/REN) was calculated by the ratio of LUC to REN. Each assay was carried out with three independent experiments, and each independent experiment was conducted with three biological replicates. Error bars are the S.E. of three biological replicates.

## 5. Conclusions

In this paper, the conjoint analysis of metabolites and transcriptional profiling of *EsMYB90* transgenic tobacco leaves confirms that the EsMYB90 transcription factor is a pivotal regulator of phenylpropanoid/flavonoid metabolism, and the ectopic expression of the *Eutrema EsMYB90* gene in tobacco plants led to a large amount of flavonoid production, which plays an important role in enhancing the antioxidative activity of plants. Moreover, consistent with the higher gene expression that operates at the early and late steps in the flavonoid biosynthesis pathway, a higher level of corresponding flavonoid metabolites were produced in the leaves of transgenic plants. This study shed new light on the transcriptional regulation of flavonoid biosynthesis, and *Eutrema EsMYB90* has great potential as a candidate gene to improve plant antioxidative capacity.

## Figures and Tables

**Figure 1 ijms-22-08751-f001:**
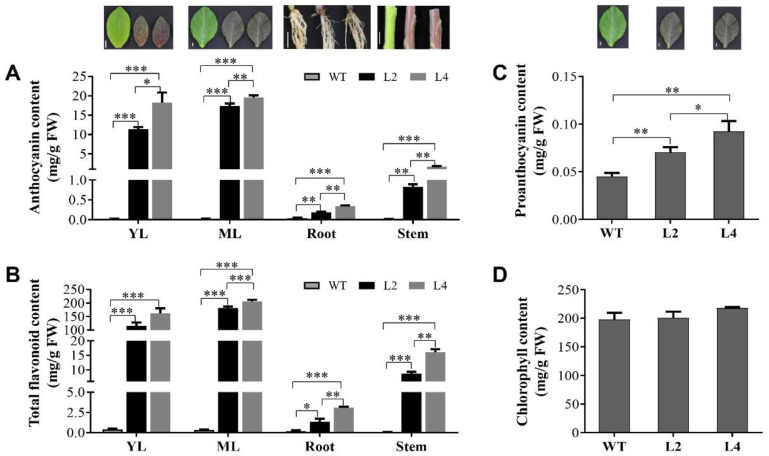
Content analysis of pigments and total flavonoids in *EsMYB90* transgenic tobacco and wild type. (**A**) Level of anthocyanin; (**B**) total flavonoid content in young leaves (YL), mature leaves (ML), roots and stems; (**C**) chlorophyll content; and (**D**) proanthocyanin content in mature leaves (ML) of transgenic line 2 (L2), line 4 (L4) and wild type (WT). The values are means ± SD of three independent biological replicates. Statistical significance: *** *p* ≤ 0.001, ** *p* ≤ 0.01, * *p* ≤ 0.05. Bar = 1 cm.

**Figure 2 ijms-22-08751-f002:**
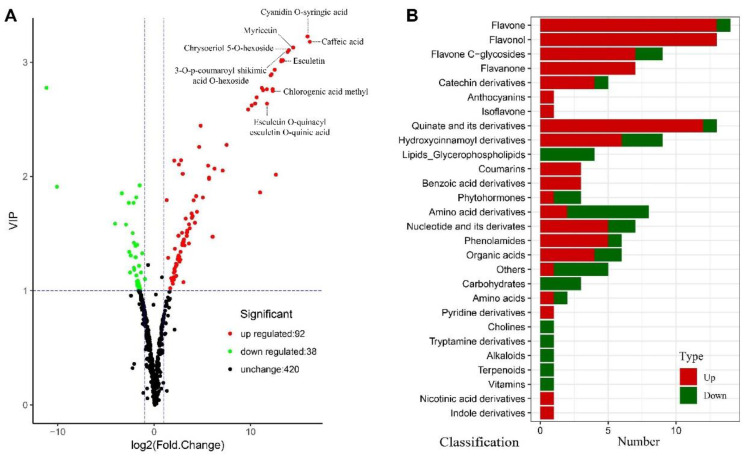
Volcano plot and classification of differential metabolites in leaves of *EsMYB90* transgenic tobacco relative to wild type. (**A**) The volcano plot exhibits metabolites detected in metabolome. The abscissa represents the logarithm of the quantitative difference multiples of metabolites (log_2_FC); the ordinate represents the variable importance in project (VIP) value. Green dots represent the down-regulated differential metabolites; red dots represent the up-regulated differential metabolites; black dots represent the metabolites that were detected with no significant difference. (**B**) Classification of the differential metabolites. The ordinate represents the name of the metabolites and the abscissa represents the number of metabolites. The red represents up-regulated metabolites and the green represents down-regulated metabolites.

**Figure 3 ijms-22-08751-f003:**
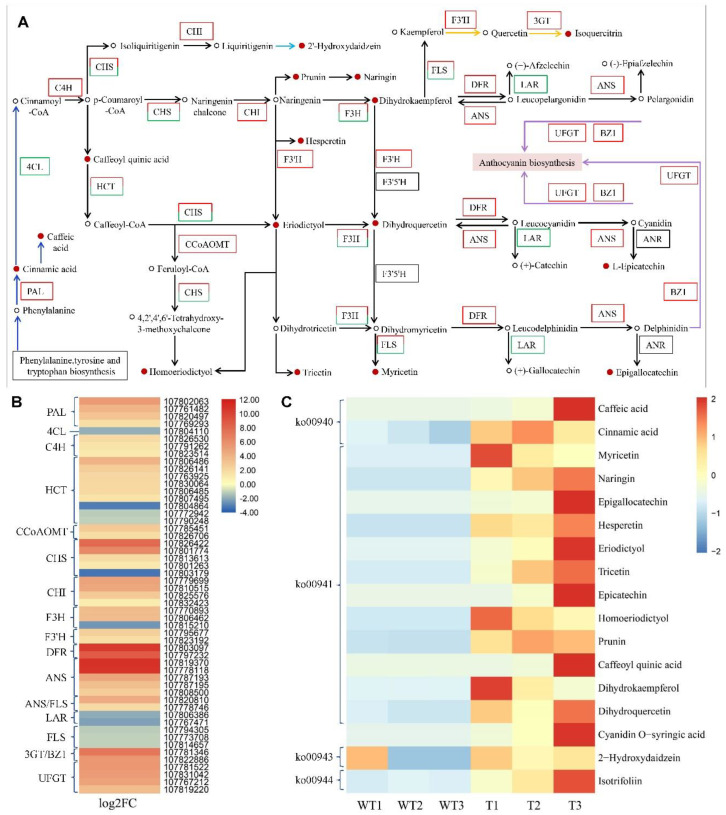
Correlation analysis of significantly differential metabolites and DEGs in flavonoid biosynthesis pathways in leaves of *EsMYB90* transgenic tobacco. (**A**) The map of integrative analysis in flavonoid metabolites and their biosynthesis-related key enzymes. Red box: significantly up-regulated DEGs encoding corresponding enzyme; green box: significantly down-regulated DEGs encoding corresponding enzyme; red/green box: significantly up/down-regulated DEGs encoding corresponding enzyme; black box: the enzyme encoding genes detected but with no significant difference. Red dot: significantly up-regulated differential metabolites; black circle: no significantly differential metabolites. Blue arrow lines: ko00940; black arrow lines: ko00941; purple arrow lines: ko00942; light blue arrow lines: ko00943; yellow arrow lines: ko00944. (**B**) Heatmap of DEGs enriched in flavonoid biosynthesis pathways. The color bar represents the logarithm of gene expression foldchange (log_2_FC), where the red indicates up-regulated genes and the blue indicates down-regulated genes. (**C**) Heatmap of differential metabolites annotated in flavonoid biosynthesis pathways. The color bar represents the level of metabolites, where the red indicates the metabolites with a higher level, and the blue indicates metabolites with a lower level.

**Figure 4 ijms-22-08751-f004:**
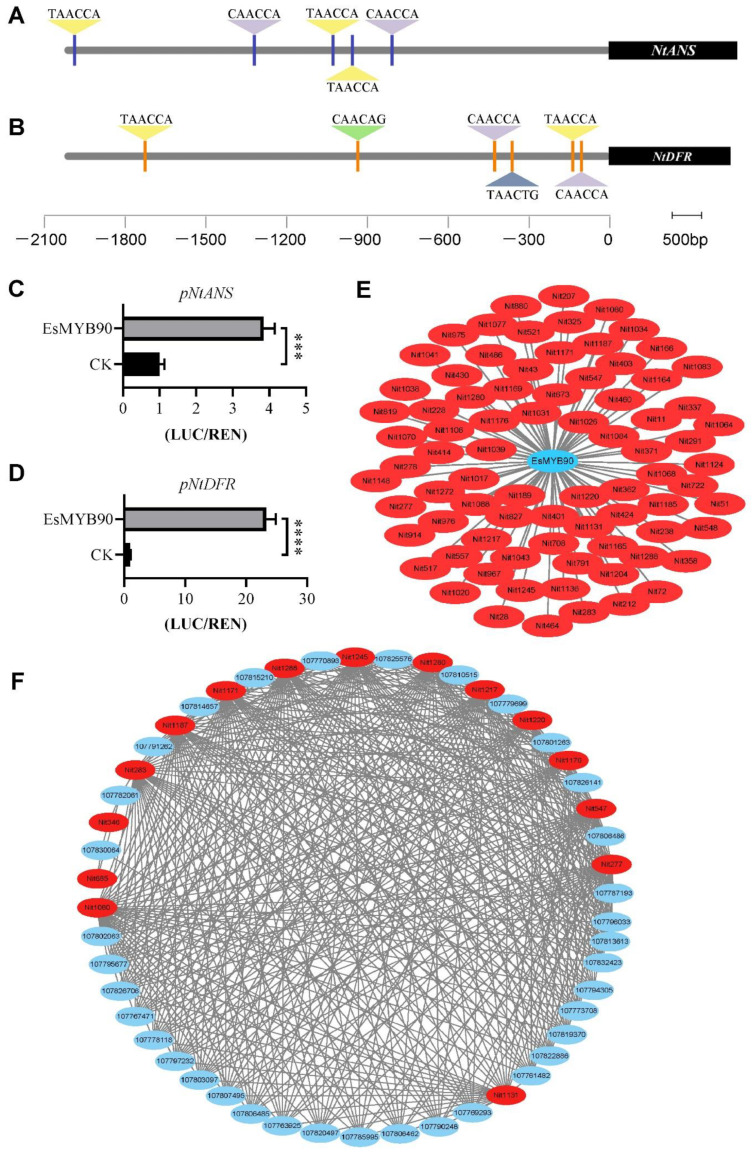
EsMYB90 activates transcription of flavonoid biosynthesis genes to modulate flavonoid metabolites in tobacco (*Nicotiana benthamiana*) leaves. (**A**) MYB-binding motifs analysis in *NtANS* promoter region. (**B**) MYB-binding motifs analysis in *NtDFR* promoter region. (**A**,**B**): Triangle indicates the predicted MYB recognized cis-elements. (**C**) Relative luciferase activity (LUC/REN) of *NtANS* promoter (*pNtANS*) by EsMYB90. (**D**) Relative luciferase activity (LUC/REN) of *NtDFR* promoter (*pNtDFR*) by EsMYB90. (**E**) Correlation network between *EsMYB90* gene and the differential metabolites. (**F**) Correlation network analysis of differential flavonoid metabolites and differential expression genes in phenylpropanoid/flavonoid biosynthesis pathways. (**E**,**F**): Red oval: significantly differential metabolites; blue oval: differential expression genes (DEGs); grey straight line indicates the correlations between the differential metabolites and DEGs. The values are means ± SD of three biological replicates. Asterisks indicate significant difference: **** *p* ≤ 0.0001, *** *p* ≤ 0.001.

**Figure 5 ijms-22-08751-f005:**
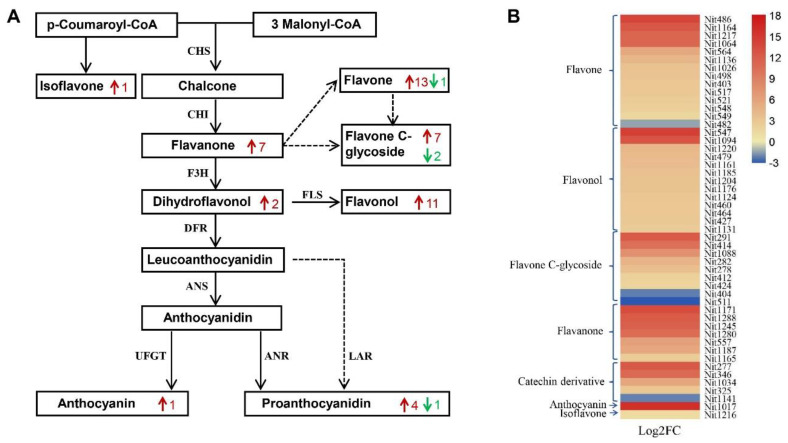
Schematic model and heatmap analysis of metabolites in flavonoid biosynthesis pathways. (**A**) Schematic model of the pathways leading to biosynthesis of anthocyanin, flavonol and PA. (**B**) Heatmap analysis of differential metabolites in flavonoid biosynthesis pathways. The color bar represents the logarithm of metabolite content foldchange (log_2_FC), where the red indicates up-regulated metabolites and the blue indicates down-regulated metabolites.

## Data Availability

All data generated or analyzed during this study are included in this published article and its supplementary files. The raw data of RNA library were available at NCBI Short Read Archive (PRJNA609528).

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
