# Peer review of "Integrated Metabolomic and Transcriptomic Analysis Reveals the Flavonoid Regulatory Network by Eutrema EsMYB90"

_ijms, 2021, doi:10.3390/ijms22168751_

Round 1
Reviewer 1 Report
The introduction should show why tobacco plants are used to conduct the trials.
In materials and methods (4.1 Plant materials and growing conditions) should be detailed the method of obtaining lines L2 and L4, what characteristics do they have and how they differ.
Information on the convenience of the experiments to be carried out is detailed in Results. This information is valuable, but it would be advisable to see it in the Materials and methods section.
In section 2.1 Increased Flavonoid Compounds in EsMYB90 Transgenic Tobacco Leaves, indicate whether or not there were significant differences in the amount of the different compounds analyzed between L2 and L4.
Figures should be of higher quality, they look blurry.
Author Response
Dear Reviewer,
We are thankful for your comments and suggestions.The following is our point-by-point response. Thanks.
Point 1: The introduction should show why tobacco plants are used to conduct the trials.
Response 1: Thank you very much for your suggestion. Tobacco, as a model plant for genetic transformation, is very convenient for transformation of genes and molecular biology research. In our previous research, we have successfully obtained EsMYB90 transgenic tobacco plants. Thus, we further carried out the combined analysis of metabolome and transcriptome of EsMYB90 transgenic tobacco plants in the present research.
Point 2: In materials and methods (4.1 Plant materials and growing conditions) should be detailed the method of obtaining lines L2 and L4, what characteristics do they have and how they differ.
Response 2: Thank you for your suggestion. In our previous research, eighteen 35S:EsMYB90 transgenic tobacco lines were obtained, and almost all transgenic tobacco lines exhibited purple-red leaves and corollas, purple-black sepals and fruit pods, whereas the leaves, sepals and fruit pods of wild-type were green, with pink corollas. Of which L2 and L4 transgenic lines displayed more obvious pigmentation (Qi, Y., et al., Identification of the Eutrema salsugineum EsMYB90 gene important for anthocyanin biosynthesis. BMC Plant Biology, 2020). In this study, the content of anthocyanin, total flavonoid and proanthocyanidin in the tissues detected of L4 EsMYB90 transgenic tobacco line were all significantly higher than that of L2 line, except for total flavonoid content in young leaves (YL). Additionally, we have added the relevant backgroud information about L2 and L4 transgenic lines in 4.1 ‘Plant materials and growing conditions’. Thanks.
Point 3: Information on the convenience of the experiments to be carried out is detailed in Results. This information is valuable, but it would be advisable to see it in the Materials and methods section.
Response 3: We are appreciated for your suggestion. In the ‘Materials and Methods’ of the paper, we have added the relevant backgroud information about L2 and L4 transgenic lines in ‘4.1.Plant materials and growing conditions’. Moreover, some more detailed method information have also added in ‘4.5.Differential Metabolite Analysis and Metabolic Pathways Construction’ and ‘4.6. Integrated Analysis of Metabolome and Transcriptome’ in the text. Thanks.
Point 4: In section 2.1 Increased Flavonoid Compounds in EsMYB90 Transgenic Tobacco Leaves, indicate whether or not there were significant differences in the amount of the different compounds analyzed between L2 and L4.
Response 4: Thank you a lot for your suggestion. We have added the significance analysis on the amount of different compounds between EsMYB90 transgenic tobacco L2 line and L4 line, and Figure 1 were also revised correspondingly. In this study, the content of anthocyanin, total flavonoid and proanthocyanidin, in the tissues detected of L4 EsMYB90 transgenic tobacco line, were all significantly higher than that of L2 line, except for total flavonoid content in young leaves (YL).
Point 5: Figures should be of higher quality, they look blurry.
Response 5: Thank you for your attention. According to the general requirement that the clarity of pictures and charts, the resolution of the Figures provided in this paper were seted as 300dpi in JPG format. In addition, we have also uploaded the Figures in PDF format for you to find a clearer version of Figures. Thanks.
Best regards
Sincerely
Quan Zhang
Reviewer 2 Report
Qi et al reported the metabolomics changes in tobacco plants overexpressing EsMYB90, a key regulator of flavonoids biosynthesis identified in a previous study. Analysis of the correlation between metabolites accumulation with expression of biosynthesis genes further prove its function as a conserved activator of flavonoids biosynthesis. They also identified two direct targets of EsMYB90 in tobacco. From applied prospect, the results suggest that EsMYB90 alone could be an useful breeding tools in certain plants.
However, the current study did not provide enough novel insights on how EsMYB90 function, instead, a descriptively support the previous discovery. There are a few remaining questions that could be addressed by the study but currently is lacking : 1. What mechanism lead cyanidin O-syringic acid as the main effect of overexpression of EsMYB90 in tobacco? 2. What are the targets of EsMYB90 in Eutrema salsugineum and could OE MYB90 improve agronomic trait? 3. Do MYB90 function as a protein complex in tobacco? If yes, what is its partner and why expression of MYB90 alone is sufficient?
Some comments on the text
- Provide both relative quantity and fold change of the given metabolites in analysis of metabolomics data.
- Line 298-324 seems more relevant to the section 2.4.
- Title:
Integrated metabolomic and transcriptomic analysis reveals the flavonoids regulatory network by Eutrema EsMYB90 and underlying regulation mechanism
- Grammar “line 46 metabolites were 19 significantly enhanced level.”?
- Figures : Explain the color bar in figure legends.
Author Response
Dear Reviewer,
We are thankful for your comments and suggestions.The following is our point-by-point response. Thanks.
Point 1: What mechanism lead cyanidin O-syringic acid as the main effect of overexpression of EsMYB90 in tobacco?
Response 1: Thank you very much for your attention. In view of anthocyanin content in the EsMYB90 transgenic tobacco L4 line were significantly up-regulated, and cyanidin O-syringic acid was the only differential metabolite detected in anthocyanin in our metabolome analysis. Thus, it is proposed that cyanidin O-syringic acid is a crucial pigment compound for the purplish red phenotype in the leaves of EsMYB90 transgenic tobacco.
As we know, the first committed reaction leading to anthocyanin and proanthocyanidin (PA) is the reduction of dihydroflavonol to the corresponding leucoanthocyanidin catalyzed by DFR. The first colored compound in the anthocyanin biosynthetic pathway is anthocyanidin, and LDOX/ANS catalyzes the formation of anthocyanidin from leucoanthocyanidin. Glycosylation of anthocyanidin catalyzed by UFGT to form anthocyanin, is essential for stable anthocyanin accumulation. In this research, all genes encoding DFR(2), ANS/LDOX (7), 3GT/BZ1(2) and UFGT (4) were markedly up-regulated (Figure 3, Supplementary Table S4), thus we inferred these up-regulated DEGs including DFR, ANS/LDOX, 3GT/BZ1 and UFGT are possibly responsible for the strongly enhanced cyanidin O-syringic acid content, in the leaves of EsMYB90 transgenic tobacco. However, cyanidin o-syringic acid has not presently been annotated into the KEGG pathway, and there is no relevant report on its research.
Point 2: What are the targets of EsMYB90 in Eutrema salsugineum and could OE MYB90 improve agronomic trait?
Response 2: Thank you a lot for your attention. In Eutrema salsugineum, the targets of EsMYB90 transcription factor should be flavonoid biosynthesis related genes, such as EsDFR, EsANS and EsUFGT genes. However, we need more experiments to clarify the EsMYB90 regulation mechanism in E.salsugineum. In the study, the dual luciferase assay have documented that EsMYB90 TF could operate as a positive transcriptional regulator of NtANS and NtDFR genes by directly binding to the MYB-binding elements of their promoters in transgenic tobacco plants.
In this study, ectopic expression of 35S: EsMYB90 in tobacco resulted in 46 flavonoids with markedly up-regulated level in transgenic tobacco (Figure 2, Supplementary Table S3). The accumulation of anthocyanin and flavonoid not only provide plants with color for most flowers, fruits, and seeds to attract pollinators, but also their lightshielding, metal-binding and antioxidant capacity and function in osmotic-regulation. Thereby, considering the enhancing the content of anthocyanin and flavonoid in food plants represents an important objective in crop genetic improvement, we have reason to believe EsMYB90 could improve agronomic trait of crop. At present, we have completed the wheat genetic transformation of EsMYB90 gene, and the content analysis of anthocyanin and other flavonoids and transcriptome analysis of transgenic wheat also proved our inference. Thanks.
Point 3: Do MYB90 function as a protein complex in tobacco? If yes, what is its partner and why expression of MYB90 alone is sufficient?
Response 3: Thank you very much for your question. In higher plants, proanthocyanidin and anthocyanin biosynthesis are regulated by different sets of MYB-bHLH-WD40 (MBW) complexes, and the R2R3-MYBs play vital roles in transcriptional regulation of anthocyanins biosynthesis. Although the genetic transformation of Eutrema EsMYB90 gene alone in tobacco is sufficient for flavonoid biosynthesis, however we speculate that EsMYB90 may form a protein complex with tobacco NtbHLH and/or NtWD40 proteins to exert its function.
Some comments on the text
Point 1: Provide both relative quantity and fold change of the given metabolites in analysis of metabolomics data.
Response 1: Thank you for your attention. We think that the relative quantity and fold change of the given metabolites in analysis of metabolomics data have been provided and showed in Table S2.
Point 2: Line 298-324 seems more relevant to the section 2.4.
Response 2: Thank you a lot for your suggestion. We can understand your thinking. However, in the section of ‘2.4. Integrated Analysis of Metabolite Profiling and RNA-seq in Phenylpropanoid/Flavonoid Biosynthesis Pathways’, the metabolites and genes introduced are only annotated in KEGG pathways of ‘Phenylpropanoid/Flavonoid Biosynthesis Pathways’ (ko00940, ko00941, ko00942, ko00943 and ko00944), except for cyanidin O-syringic acid.
In the section of ‘2.5. EsMYB90 Enhanced Flavonoid metabolites Level via Activating Transcription of Flavonoid Biosynthesis Genes’, the metabolites introduced were the ‘Enhanced Phenylpropanoid/Flavonoid metabolites regulated by EsMYB90’ that not only were the metabolites annotated in KEGG pathways of ‘Phenylpropanoid/Flavonoid Biosynthesis Pathways’. Thanks.
Point 3: Title: Integrated metabolomic and transcriptomic analysis reveals the flavonoids regulatory network by Eutrema EsMYB90 and underlying regulation mechanism
Response 3: We appreciate your suggestion. At present, we have revised the title as ‘Integrated metabolomic and transcriptomic analysis reveals the flavonoids regulatory network by Eutrema EsMYB90’. But we wonder if the editor could agree with the amendment of title. Thanks.
Point 4: Grammar “line 46 metabolites were 19 significantly enhanced level.”?
Response 4: Thank you very much for your suggestion. We think that you mean the sentence ‘A total of 50 markedly differential flavonoids including flavones (14), flavonols (13), flavone C-glycosides (9), flavanones (7), catechin derivatives (5), anthocyanins (1) and isoflavone (1) were identified, and 46 metabolites were significantly enhanced level’ in the ‘Abstract’. Herein, we have revised the grammar as ‘……, of which 46 metabolites were with significantly enhanced level’ in the text. Thanks.
Point 5: Figures : Explain the color bar in figure legends.
Response 5: Thank you for your suggestion. We have added explanations of the color bar in Figure 3 and Figure 5. Figure 3: (B) Heatmap of DEGs enriched in flavonoid biosynthesis pathways. The color bar indicates the logarithm of gene expression foldchange (log2FC), and the red indicates up-regulated genes and the blue indicates down-regulated genes. Figure 3: (C) Heatmap of differential metabolite annotated in flavonoid biosynthesis pathways. The color bar represents the level of metabolites, and the red indicates the metabolites with the higher level, and the blue indicates metabolites with the lower level. Figure 5: (B) Heatmap analysis of differential metabolites in flavonoid biosynthesis pathways. The color bar represents the logarithm of metabolites content foldchange (log2FC), and the red indicates up-regulated metabolites and the blue indicates down-regulated metabolites.
Best regards
Sincerely
Quan Zhang
Reviewer 3 Report
- In my opinion, it is well-designed and well-presented research. I have marked minor mistakes on pdf.
- Authors should carefully check the entire paper if italic or regular font is used correctly (gene/protein names).
- All text needs to be understandable for all readers. The abbreviations list should include MYB or KEGG, too.
- The abbreviations list should be in alphabetical order.
- Table S2. The number of signs presented after a coma should be reduced. It is impossible to detect it so precisely.
- Table S1 is cited in the last paragraph (line 531). Supplemented tables should be renumbered in order of appearance in the text.
Author Response
Dear Reviewer,
We are thankful for your comments and suggestions.The following is our point-by-point response. Thanks.
Point 1: In my opinion, it is well-designed and well-presented research. I have marked minor mistakes on pdf.
Response 1: Thank you very much for your suggestion. We have made the corresponding amendment in the text. Thanks.
Point 2: Authors should carefully check the entire paper if italic or regular font is used correctly (gene/protein names).
Response 2: We are grateful for your suggestion. We have carefully checked and revised the italic or regular font in the text. The revised italic or regular font can be seen as the revised format in the text. Thanks.
Point 3: All text needs to be understandable for all readers. The abbreviations list should include MYB or KEGG, too.
Response 3: We are appreciated for your suggestion. We have added the abbreviations of MYB (v-myb avian myeloblastosis viral oncogene homolog) and KEGG (Kyoto Encyclopedia of Genes and Genomes ) in abbreviations list of the text.
Point 4: The abbreviations list should be in alphabetical order.
Response 4: We have made the amendment. Thanks.
Point 5: Table S2. The number of signs presented after a coma should be reduced. It is impossible to detect it so precisely.
Response 5: Thank you a lot for your suggestion. We have reduced the number of decimal places in the digits in table S1 (previously named Table S2) .
Point 6: Table S1 is cited in the last paragraph (line 531). Supplemented tables should be renumbered in order of appearance in the text.
Response 6: Thank you a lot for your suggestion. We have renumbered the supplementary tables.
Best regards
Sincerely
Quan Zhang
Round 2
Reviewer 2 Report
The authors have addressed all my questions. I encourage the authors to consider discussing functional conservation of the EsMYB90 as a perspective in the text, which may provide insights for application of EsMYB90 in different crops.
Author Response
Dear Reviewer,
We are thankful for your attention.The following is our response to your suggestion. Thanks.
Point: The authors have addressed all my questions. I encourage the authors to consider discussing functional conservation of the EsMYB90 as a perspective in the text, which may provide insights for application of EsMYB90 in different crops.
Response: Thank you for your suggestions.The following is the revised and added contents in the discussion section of the article.
3.3. EsMYB90 an excellent gene for genetic breeding to improve agronomic trait of crops
In recent several decades, the majority of flavonoid molecules, and the genes involved in biosynthesis of these diverse compounds in Arabidopsis, have been intensively identified [53]. Moreover, a huge number of genetic engineering attempts have been described to produce novel flower colors in several plant species, such as petunia, gerbera, rose and carnation by modifying the anthocyanin biosynthesis pathway [1-3]. However, the diversified functionality of various flavonoids and the transcription regulation of the enormous diversification of flavonoids should be addressed by further investigation.
In higher plants, the flavonoid and anthocyanin biosynthesis are regulated by conserved MYB-bHLH-WD40 (MBW) complex, and the different sets of MBW complexes exert in different plants [2]. Of which, R2R3-MYB play pivotal role in transcriptional regulation of flavonoid biosynthesis. EsMYB90 protein, a R2R3-MYB from Eutrema, has a conserved DNA-binding domain (R2 and R3 repeats) in N-terminal, and a conserved [D/E]Lx2[R/K]x3Lx6Lx3R motif required for interaction with R/B-like bHLH proteins[37, 57]. The phylogenetic tree of EsMYB90 with 29 MYB proteins involved in proanthocyanidin and anthocyanin synthesis in 16 plants demonstrated that EsMYB90 has the closer relationship with Arabidopsis AtMYB75, AtMYB90, AtMYB113, AtMYB114, Brassica oleracea BoMYB1 and Brassica rapa BrMYB114 [37]. Further, ANDV motif, a characteristic identifier for anthocyanin-promoting MYBs in dicots, which existed in EsMYB90, AtMYB90, AtMYB75, AtMYB113, AtMYB114 [37, 57]. However, EsMYB90 protein exist only 80.5, 78.9, 78.4, 74.4, 65.9, 50% identities respectively to BoMYB1, AtMYB90, BrMYB114, AtMYB75, AtMYB113 and AtMYB114 [37]. In our current findings, the direct activation of NtANS and NtDFR exerted by EsMYB90 protein, which resulted in phenotypic effect of anthocyanin accumulation in transgenic tobacco plants. Furthermore, we demonstrated that the critical regulation function of Eutrema EsMYB90 gene on flavonoids regulatory network that allowed the significant accumulation of 46 flavonoid compounds. These findings could be the basis for further engineering of flavonoids and optimization of the metabolic pathway.
Biotechnological application for the promotion of human health by engineering of flavonoid is a promising area [58, 59]. Considering various flavonoids not only providing the abundant colors to plants, but also possessing extensive antioxidative effects, antiviral activities and neuroprotective properties, thereby their accumulation is a key objective for the genetic improvement of crops [3, 60, 61]. In summary, EsMYB90 transcription factor on the regulatory network involved in several already annotated flavonod biosynthesis genes and flavonoid metabolites, revealed that EsMYB90 gene is an excellent gene for genetic breeding to improve agricultural and economic performance of crops, and the study of modifying the flavonoids content by Eutrema EsMYB90 in plant tissues also opens up a new avenue to promote the commercial value of the crops.
Best regards
Sincerely
Quan Zhang